# Pregnancy and Childbirth in Uterus Didelphys: A Report of Three Cases

**DOI:** 10.3390/medicina56040198

**Published:** 2020-04-23

**Authors:** Stanislav Slavchev, Stoyan Kostov, Angel Yordanov

**Affiliations:** 1Department of Gynecology, Medical University Varna, 9000 Varna, Bulgaria; st_slavchev@abv.bg (S.S.); drstoqn.kostov@gmail.com (S.K.); 2Medical University–Varna, 9000 Varna, Bulgaria; 3Department of Gynecologic Oncology, Medical University Pleven, 5800 Pleven, Bulgaria

**Keywords:** congenital anomalies of the Müllerian ducts, uterus didelphys, pregnancy, outcome

## Abstract

Uterus didelphys is a rare form of congenital anomaly of the Müllerian ducts. The clinical significance of this anomaly of the female reproductive tract is associated with various reproductive issues: increased risk of preterm birth before 37 weeks’ gestation, abnormal fetal presentation, delivery by caesarean section, intrauterine fetal growth restriction, low birth weight less than 2500 g, and perinatal mortality. We present three cases of uterus didelphys and full-term pregnancy, which resulted in favorable birth outcomes of live-born, full-term infants. In two of the cases, delivery was performed via Caesarean section: due to lack of labor activity in one of the cases and lack of response to oxytocin stimulation in the second case. The weight of two of the new-born infants was lower than expected for the gestational age.

## 1. Introduction

Congenital anomalies of the female reproductive tract are of special interest because of their association with various reproductive difficulties: impaired possibility of natural or assisted conception, increased rate of first and second trimester miscarriages, preterm birth, placental abruption, lower birth weight and fetal growth restriction, malpresentation at delivery, and perinatal mortality [1]. The prevalence of congenital uterine anomalies in the general population is 5.5%, 8.0% in women with infertility, 13.3% of the population with abortions, and reaches 24.5% in patients with abortions and infertility [2]. Congenital malformations of the female genital tract represent a heterogeneous group and have their origin in the abnormal formation, confluence, or resorption of the Müllerian ducts during fetal development [3]. Various congenital anomalies are specifically related to the female reproductive problems in different ways and to different extents. The most severe disorders have the most significant impact [4]. Currently, there are various classification systems for the categorization of congenital reproductive tract malformations. The oldest and most commonly used classification is that of 1988 of the American Society for Reproductive Medicine (ASRM, formerly the American Fertility Society). The ASRM classification divides Müllerian duct anomalies into seven major types according to the anatomical changes in the uterus and the embryonic processes responsible for them. However, it does not account for complex urogenital malformations [5]. In 2013, the European Society of Human Reproduction and Embryology (ESHRE) and the European Society for Gynaecological Endoscopy (ESGE) published a classification of female genital anomalies. It is designed and developed primarily on the basis of anatomical findings. Anomalies are classified into main classes and sub-classes, reflecting separately anatomical abnormalities and variations; uterine, cervical, and vaginal anomalies are classified independently into sub-classes [6].

The ASRM classification defines uterus didelphys, or didelphic uterus, as a reproductive tract anomaly representing a complete duplication of the uterus and cervix. It occurs as a result of a failure of Müllerian duct fusion in 8 weeks’ gestation [5,7] The ESHRE/ESGE classification system identifies this congenital anomaly as U3b/C2 (complete bicorporal uterus/double “normal” cervix) (Table 1) [6].

Generally, this anomaly is presented by two uterine cervixes and two uteri, each linked to a fallopian tube. A longitudinal vaginal septum is present in 75% of the cases. Duplication in other organs-vulva, bladder, urethra, and anus may also be observed [7].

## 2. Case Reports

We present three cases of the anomaly uterus didelphys at a full-term pregnancy and the difficulties faced in our attempt to achieve normal delivery.

### 2.1. Case 1

A 21 year-old patient was diagnosed with uterus didelphys anomaly one year earlier in the course of an undeveloped twin pregnancy in each uterine cavity. A residual abrasion was performed separately for both uteri at 8 weeks’ gestation. During her second pregnancy follow-up, the fetal estimated weight obtained by ultrasonography at 37 weeks was 2951g (between 50th–75th centile) [8]. The fetus was female. The amniotic fluid index (AFI) was above the 5th centile and the umbilical artery Doppler’s was normal. The patient was hospitalized in the maternity ward at 38 weeks’ gestation without having any contractions, with a spontaneous rupture of membranes. Gynaecological examination revealed a longitudinal septum separating the vagina and two cervixes (Figure 1).

There was no dilatation of the right cervix, the left cervix was dilated to 2 cm with clear amniotic fluid leakage, and a Bishop score of 4 points (cervical position—middle, cervical consistency—medium, cervical effacement—50%, cervical dilatation—2 cm, fetal station—3) [9]. External pelvimetry revealed a normal size pelvis. An ultrasound examination revealed a living fetus in the left uterus and occipital-iliac dextra anterior (OIDA) fetal presentation. Cardiotocography monitoring showed a baseline fetal heart rate of 140 beats per minute with preserved variability and fetal reactivity. Sixteen hours after admission irregular birth activity was registered with single short-term contractions at 7–8 min intervals. Oxytocin stimulation resulted in no change in obstetric status for about 5 h (Bishop score—5, cervical effacement—60–70%), and delivery continued via Caesarean section. Uterus didelphys was confirmed during surgery: two uteri, each with one fallopian tube and an ovary, were clearly defined (Figure 2 and Figure 3).

A live-born infant weighing 2580 g, height 48 cm, and with an Apgar score of 9–10 points was delivered. The patient postpartum time period was uncomplicated. On the follow-up visit between 4–6 weeks, the patient was in good health condition and without complains.

### 2.2. Case 2

A 25 year-old first-time pregnant woman was admitted to the maternity ward 6 days after the likely probable expected date of delivery with an intact amniotic membrane and no labor activity. Uterus didelphys anomaly was diagnosed using 2D ultrasound examination on an outpatient basis during a female consultation visit. After spontaneous conception, ultrasonography at 8 weeks of gestations was performed (Figure 4).

The fetal estimated weight obtained by ultrasonography during patient pregnancy consultation at 37 weeks was 3143 g (between 50th–75th centile) [8]. The fetus was male. The amniotic fluid index (AFI) was above the 5th centile and the umbilical artery Doppler’s were normal. The obstetric examination after admission revealed a longitudinal vaginal septum, two 80% shortened uterine cervixes with 0.5 cm widening, and a Bishop score of 4 points (cervical position–posterior, cervical consistency–medium, cervical effacement—80%, cervical dilatation—0.5 cm, fetal station—3). The external pelvimetry showed normal pelvic dimensions. An ultrasound examination confirmed pregnancy in the left uterus. Fetal presentation was defined as occipital-iliac sinister anterior. Cardiotocography showed no deviations from the norm. Induction of oxytocin was performed for the duration of two days to induce labor activity. Due to the lack of effect and any change in obstetric status (Bishop score—4), the decision to proceed with a C-section was made. The genital uterine anomaly, i.e., uterus didelphys, was verified during surgery. A live-born male infant of 2730 g birth weight, 48 cm height, and Apgar score of 9–10 points was delivered. The patient postpartum time period was uncomplicated. On the follow-up visit between 4-6 weeks, the patient was in good health condition and without complains.

### 2.3. Case 3

A 27 year-old patient at the 38th week of her third pregnancy, with initial labor activity, cardiotocographically verified contractions 5 min apart. The patient’s first pregnancy had ended as a spontaneous abortion at the 7th week. Her second pregnancy outcome was a normal delivery to a live-born infant weighing 3000 g. During her third pregnancy follow-up, the fetal estimated weight obtained by ultrasonography at 36 weeks was 2713 g (between 25th–50th centile) [8]. The amniotic fluid index (AFI) was above the 5th centile and the umbilical artery Doppler’s were normal. Examination at admission revealed uterus didelphys, the presence of longitudinal vaginal septum and two cervixes: the right cervix was with medium cervical consistency and 3 cm dilated, the Bishop score was 6 points (cervical position—middle, cervical effacement—50%, fetal station—−2). Pelvic measurements were normal. Fetal position was defined as occipital-iliac sinister anterior. Eight hours after admission to the maternity ward, there was no change in the obstetric status and no cardiotocographically registered contractions. Eleven hours later, contractions at intervals of 7 min and 1 cm additional cervix dilation were registered. After the initiation of oxytocin stimulation, in another 5 h, the patient gave birth to a normal full-term live-born male infant weighing 3250 g, a height of 53 cm, and an Apgar score of 9–10 points. The vaginal septum was pushed by the presenting fore-part of the fetus and did not develop complications during labor. On the second day of the postpartum time period, the patient experienced abdominal pain in the ascending colon, which required consultation with a surgeon. The diagnosis of acute surgical abdomen was rejected, treatment with non-steroidal anti-inflammatory drugs was performed and the symptoms resolved. On the follow-up visit between 4–6 weeks, the ultrasound and vaginal examination were normal. We summarize the information of the three case reports in Table 2.

## 3. Discussion

Uterus didelphys is a rare anomaly and accounts for 8% of the congenital anomalies of the female reproductive tract [10]. It occurs in 0.3% of the total population. In the population of women with a history of abortion and infertility, its rate of occurrence is more frequent in 2.1% [2]. Most often, this anomaly is asymptomatic and accidentally detected which is likely the reason for the inaccurate assessment of its frequency. Its clinical manifestations may present with pelvic discomfort, dyspareunia, dysmenorrhea, hematocolpos, and hematometra [11]. Although the initial diagnosis of uterus didelphys is made by ultrasound or hysterosalpingography, MRI offers the most accurate diagnosis. MRI is the best option for the classification of the various anomalies because of its better anatomic assessment compared with other diagnostic modalities [12]. Accurate diagnosis is essential to determine the most effective treatment during childbirth.

The ability to conceive is not typically impaired, but pregnancy in uterus didelphys is often associated with reproductive failure. A meta-analysis of 25 studies involving more than 160,000 women, of which 3766 with congenital uterine anomalies, shows that uterus didelphys was not associated with decreased natural or assisted fertility and an increased frequency of spontaneous abortions. The research establishes an increased risk of preterm birth before 37 weeks’ gestation, breech presentation, intrauterine growth retardation in the fetus, low birth weight infants of less than 2500 g, and perinatal death [1].

Concerning the presented clinical cases, feto-pelvic-incompatibility and fetal asphyxia were excluded in all three patients, thus proceeding to a normal delivery. The administration of oxytocin stimulation or oxytocin induction was required in all three patients, as the inadequate response of uterine contractions was observed. In the first case, labor stimulation led to irregular and weak uterine contractions. In the second case, induction was unsuccessful. With both patients we proceeded with a surgical delivery. In the third patient, medications administered stimulation resulted in a normal childbirth but with a protracted first period. More than 80% of the pregnancies and childbirths in women with uterus didelphys end with a caesarean section but without immediate indication for it, research literature shows [13]. Vaginal septum may cause dystocia of the soft birth canal [13]. Other research cases concluded that patients with congenital anomalies of the Müllerian ducts have a high Caesarean section (CS) rate (53%) which is highest (82%) in the uterus didelphys group [14,15]. Pankaja et al. reported three cases with uterus didelphys and their pregnancy outcomes. CS was performed in two cases due to failure of the labor to progress and previous uterine surgery. The authors concluded that uterus didelphys is related with high incidence of CS due to labor dystocia [14]. Zhang et al. in a retrospective study evaluated the fertility and obstetric outcome of 116 inpatients with uterine malformations and pregnancy. Authors observed significantly higher rates of CS (78.5%) in the group of uterine anomalies compared with the group of women with a normal uterus [16]. Wondimu and Barsenga reported a case with uterus didelphys and term pregnancy. CS was performed due to soft tissue dystocia. They concluded that Uterus didelphys should be considered in the intrapartum evaluation of women with suspected soft tissue dystocia [17]. Othman studied a population of 286 pregnant women diagnosed with uterus didelphys. There were an increased rate of deliveries by CS (80.8%) compared to spontaneous vaginal deliveries (5.9%) [18]. In the case in which we have achieved a normal childbirth (case 3), no ligation of the vaginal septum was required as it did not interfere with normal delivery. However, in the same patient who was admitted with initial labor activity, we observed a case of uterine contractions for the period of 11 h. The main reason for performing CS in a patient with uterine didelphys is the breech presentation or dystocia as a result of the obstruction of the pelvic entrance by the non-pregnant uterine cavity [13]. Notwithstanding that labor dystocia in women with uterus didelphys may result from mechanical impediment, the question arises as to whether abnormal embryonic development of the reproductive organs is not a cause for the impaired function of the myometrium.

On the basis of our review of the literature, we think that CS and uterus didelphys is a sufficient, but not absolute indication for CS. However, although uterus didelphys and normal deliveries have been reported even after CS, we believe that surgical delivery is a safer method for this Müllerian anomaly [19]. A more complex analysis of the accumulated scientific data is needed to determine the optimal mode of delivery for this anomaly and to assess the risk of natural childbirth for the fetus or the mother.

## 4. Conclusions

Congenital Müllerian anomalies are challenging case scenarios for obstetrician-gynecologists in terms of diagnosis and resolution of reproductive problems. Women with uterine anomalies are more likely to experience adverse pregnancy outcomes, which requires accurate knowledge, early diagnosis, and adequate treatment.

## Figures and Tables

**Figure 1 medicina-56-00198-f001:**
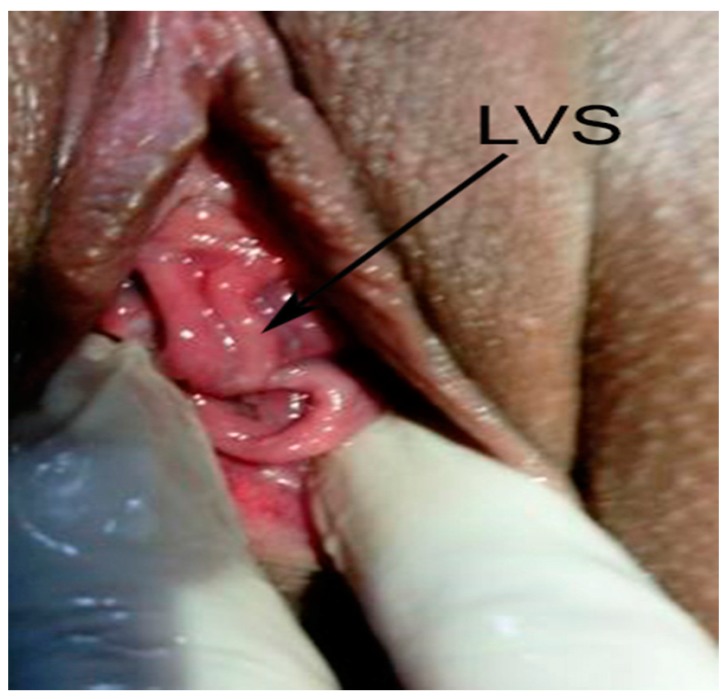
LVS—longitudinal vaginal septum.

**Figure 2 medicina-56-00198-f002:**
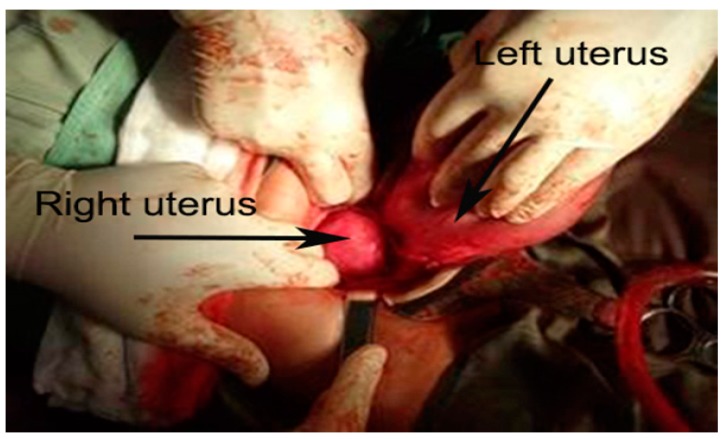
Uterus didelphys–the fetus was in the left uterine cavity, no pregnancy in the right uterine cavity.

**Figure 3 medicina-56-00198-f003:**
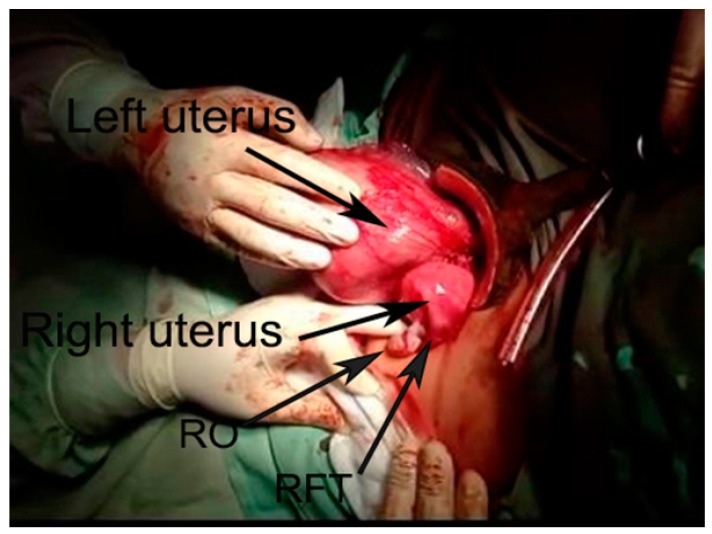
Two uteri with a fallopian tube and an ovary originating from each one. In this case, in the left uterus was the fetus. RO—right ovary, RFT—right fallopian tube.

**Figure 4 medicina-56-00198-f004:**
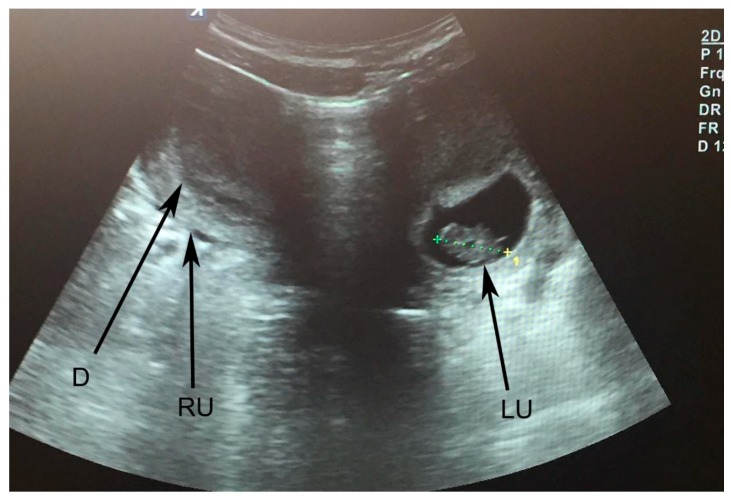
Uterus didelphys. LU—left uterus; RU—right uterus; D—decidualization of right uterine cavity; CRL is measured—8 weeks of gestation in the left uterus.

**Table 1 medicina-56-00198-t001:** European Society of Human Reproduction and Embryology (ESHRE)/Society for Gynaecological Endoscopy (ESGE) classification. Uterus didelphys is classified as U3b/C2.

Uterine Anomaly		Cervical/Vaginal Anomaly
Main Class	Main Sub-Class	Co-Existent Class
Class 0 Normal uterus Class I Dysmorphic uterus Class II Septate uterus Class III Bicorporeal uterus Class IV Hemi-uterus Class V Aplastic Class VI Unclassified malformations	a. T-shaped b. Infantilis c. Others a. Partial b. Complete a. Partial b. Complete c. Bicorporeal septate a. With rudimentary cavity (communicating or not horn) b. Without rudimentary cavity (horn without cavity/no horn) a. With rudimentary cavity (bi- or unilateral horn) b. Without rudimentary cavity (bi- or unilateral uterine remnants/aplasia)	C0: Normal cervix C1: Septate cervix C2: Double ‘normal’ cervix C3: Unilateral cervical aplasia C4: Cervical aplasia V0: Normal vagina V1: Longitudinal non-obstructing vaginal septum V2: Longitudinal obstructing vaginal septum V3: Transverse vaginal septum and/or imperforate hymen V4: Vaginal aplasia

**Table 2 medicina-56-00198-t002:** Uterus didelphys–prenatal and postnatal details of three case reports.

	Case 1	Case 2	Case 3
Maternal age at booking	21	25	27
Parity	Primipara	Primipara	Multipara
Prenatal complications	No	No	No
Prenatal USG	Pregnancy at left uterus; no kidney anomalies	Pregnancy at left uterus; no kidney anomalies	Pregnancy at right uterus; no kidney anomalies
Vaginal septum	Longitudinal	No	Longitudinal
Miscarriages	First pregnancy-missed abortion at two uterine cavities	No	First pregnancy-spontaneous abortion at 7 weeks
Gestation at delivery	38 weeks	39 weeks	38 weeks
Type of delivery	Emergency CS	Emergency CS	Vaginal
Malpresentation at delivery	Cephalic	Cephalic	Cephalic
Birth weight	2580 g	2730 g	3250 g
Apgar score	9–10	9–10	9–10
Complications at postpartum period	No	No	Pain in the ascending colon, No acute abdomen

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
