# Peer review of "Pregnancy and Childbirth in Uterus Didelphys: A Report of Three Cases"

_medicina, 2020, doi:10.3390/medicina56040198_

Round 1

Reviewer 1 Report

-the authors should include the complete classification of congenital anomalies of the Müllerian ducts, pointing out the place of uterus didelphys (ASRM or ESHRE)

-if it is possible to insert ultrasound pictures from the first trimester of pregnancy ( or before pregnancy, or even at admission because the authors specified that they performed ultrasound at admission), showing the didelphys uterus 

-also, it is important to talk about the follow-up, also at the vaginal birth and c-section patients

-in the discussion section, I would recommend discussing these cases comparative with other studies regarding the mode of delivery, in order to conclude if these anomalies are a strong indication for c section

-the authors should respect the manuscript preparation, reporting introduction, case reports  discussion and conclusions (in this draft introduction and the case reports are in the same template

-if it is possible, the new  references inserted should be more up to date ( from 2010 until nowadays)

Reviewer 2 Report

The authors of this manuscript aim to describe three cases of uterus didelphys and full-term pregnancy, which resulted in favorable birth outcomes of live-born full-term infants. The topic is of moderate interest today. The manuscript is well written. Point of strength is the figures. Two major comments should be addressed to improve the contents of the manuscript:

Major comments:

-  The authors reported that the weight of two new-born was lower than that expected for gestational age. Could the authors provide some data about fetal estimated weight obtained in the third semester ultrasound? Was the low weighted already suspected at previous ultrasonographic scans during pregnancy?

- The authors should report a standardized score (i.e. Bishop, modified Bishop score) for describing obstetrical examinations at hospital admission and during the labor.

Minor comments:

- I would separate the three cases in three different subparagraphs

-The legends to figures should be improved as at the moment they are not such informative

-The size of the figures should be increased by 120-130%

- Reference [10] is related to three evaluation of uterine malformation. This reference is too old, and at the moment new original researches or reviews on this topic have been published.

Round 2

Reviewer 2 Report

The authors adequately improved the manuscript, following reviewers' suggestions. However, a have two minor comments for the manuscript.

Please insert the estimated fetal weight (EFW) centiles for gestational age, as the fetal weight as absolute measure is not such informative (as you correctly done for AFI scores).

The tables and the figures should be adequately formatted following the journal standards.

Author Response

Dear reviewer

 We made changes as you recommended. We incorrporated the information about  fetal weight centil in the manuscript. We formatted the tables and pictures.